Physiological and anatomical investigation of the auditory brainstem in the Fat-tailed dunnart (Sminthopsis crassicaudata)

Garrett Andrew andrew-richard.garrett@biologie.uni-goettingen.de andrew.garrett.de@gmail.com 1 2
Lannigan Virginia 2 3
Yates Nathanael J. 3 4
Rodger Jennifer 2 3
Mulders Wilhelmina 3 5
1 Department of Systems Neuroscience, J.F.B. Institute für Zoologie und Anthropologie, Universität Göttingen , Germany
2 School of Biological Sciences, University of Western Australia , Crawley , Western Australia , Australia
3 School of Human Sciences, M311, University of Western Australia , Crawley , Western Australia , Australia
4 The Queensland Brain Institute, University of Queensland , St Lucia , Queensland , Australia
5 Ear Science Institute Australia , Subiaco , Western Australia , Australia
Reser David
Electronic publication date: 2019 Sep 30
Publication date: 2019
Volume: 7
Electronic Location ID: e7773
Received 2019 May 31; Accepted 2019 Aug 27
Copyright: ©2019 Garrett et al.
Copyright year: 2019
Copyright holder: Garrett et al.
License: This is an open access article distributed under the terms of the Creative Commons Attribution License, which permits unrestricted use, distribution, reproduction and adaptation in any medium and for any purpose provided that it is properly attributed. For attribution, the original author(s), title, publication source (PeerJ) and either DOI or URL of the article must be cited.
License URL: https://creativecommons.org/licenses/by/4.0/

Keywords: Superior olivary nuclei, Cochlear nucleus, Marsupial, Hearing, Auditory brainstem response

Funding: NHMRC fellowship APP1002258 Raine Fellowship Jennifer Rodger was supported by a NHMRC fellowship (APP1002258) and a Raine Fellowship. The funders had no role in study design, data collection and analysis, decision to publish, or preparation of the manuscript.

==============================
The fat-tailed dunnart (Sminthopsis crassicaudata) is a small (10–20 g) native marsupial endemic to the south west of Western Australia. Currently little is known about the auditory capabilities of the dunnart, and of marsupials in general. Consequently, this study sought to investigate several electrophysiological and anatomical properties of the dunnart auditory system. Auditory brainstem responses (ABR) were recorded to brief (5 ms) tone pips at a range of frequencies (4–47.5 kHz) and intensities to determine auditory brainstem thresholds. The dunnart ABR displayed multiple distinct peaks at all test frequencies, similar to other mammalian species. ABR showed the dunnart is most sensitive to higher frequencies increasing up to 47.5 kHz. Morphological observations (Nissl stain) revealed that the auditory structures thought to contribute to the first peaks of the ABR were all distinguishable in the dunnart. Structures identified include the dorsal and ventral subdivisions of the cochlear nucleus, including a cochlear nerve root nucleus as well as several distinct nuclei in the superior olivary complex, such as the medial nucleus of the trapezoid body, lateral superior olive and medial superior olive. This study is the first to show functional and anatomical aspects of the lower part of the auditory system in the Fat-tailed dunnart.

Introduction

Marsupials evolved separately from eutherian mammals in the Cretaceous period and now form a highly diverse group with populations in the Americas and Australia (Luo et al., 2011; Nilsson et al., 2010). One marsupial, the fat-tailed dunnart (Sminthopsis crassicaudata), is a small (10–20 g) insectivorous Australian marsupial (Frey, 1991; Morton, 1978a) that is named after its characteristic swollen tail that contains stored fat (Godfrey, 1968). The fat-tailed dunnart is a solitary animal with a widespread distribution across the southern and western parts of Australia inhabiting a variety of arid environments including open woodland, low scrublands, grasslands on clay or sand soils and farmlands (Morton, 1978a). Within these varied environments, the nocturnal dunnart hunts predominantly insects while itself being preyed upon by other predators such as snakes, feral cats and barn owls (Morton, 1978b).

Interestingly, the visual system in the fat-tailed dunnart has been shown to be different from most other marsupials as well as most eutherian mammals as they are trichromatic (Cowing et al., 2008; Ebeling, Natoli & Hemmi, 2010). Being predominantly nocturnal (Levy et al., 2019) the fat-tailed dunnart is likely to also heavily depend on its sense of hearing and its ability to localise sound as a means for prey detection, predator avoidance and species-specific communication (Osugi et al., 2011). Previous work in a range of marsupial families such as northern quoll (Dasyurus hallucatus) (Aitkin, Nelson & Shepherd, 1994; Aitkin, Nelson & Shepherd, 1996), brush-tailed possums (Trichosurus vulpecula) (Signal, Foster & Temple, 2001), and the tammar wallaby (Macropus eugenii) (Liu, 2003; Liu, Hill & Mark, 2001) has shown that the overall structure of the auditory brainstem is largely consistent with eutherian mammals, enabling the distinction of several subnuclei in cochlear nuclei (CN), superior olivary complex (SOC) and inferior colliculus (Aitkin, 1998).

However, the relative size and detailed structure of the subcortical structures in the auditory system is known to be highly varied both in eutherian mammals and marsupials (Glendenning & Masterton, 1998). For example, the CN represents about 13% of the whole auditory system in the swamp wallaby, but approximately 37% in the pocket gopher. In addition, there exists a large degree of heterogeneity in the anatomical architecture of the CN and principal nuclei of the SOC (Glendenning & Masterton, 1998). For example, in some of the Muridae such as rat, mouse and gerbil (López et al., 1993) as well as in some marsupials (Willard, 1993) the auditory nerve contains a small group of large neurons, the so-called cochlear nerve root neurons, whereas this does not appear to be the case in for instance cat or guinea pig. In the SOC, the lateral superior olive (LSO) forms a S-shaped segment in many species such as guinea pig, cat and gerbil (Grothe & Park, 2000) but has been described as a triangle shape in marsupials (Aitkin, 1996). In marsupials the cochlear nucleus is located medial to the restiform body, whereas in other mammals the it is found lateral to the restiform body (Aitkin, 1996).

With regard to functional studies, the auditory brainstem response (ABR) has been shown to reveal the typical waveforms (i.e., waves I-V present) between 1–90 kHz with lowest thresholds between 12–16 kHz in the short-tailed opossum (Monodelphus domestica) (Reimer, 1996). Click-evoked ABRs obtained from tammar wallaby also showed typical peaks and the appearance of the peaks during development correlated with the development of the known anatomical substrates of the ABR waves (Liu, 2003; Liu, Hill & Mark, 2001).

With the exception of a few references to the stripe-faced dunnart (Sminthopsis macroura) by Aitkin (1998) very little is known about the anatomy and physiology of the dunnart auditory system. In view of the fact that the fat-tailed dunnart has specific adaptations in its visual system, this paper explored functional and anatomical aspects of its auditory system to investigate whether this sensory system also has distinct features compared to other marsupials. For this purpose, we combined electrophysiological (ABR) and anatomical (Nissl staining) investigations of the auditory brainstem in the dunnart. For the latter we focussed on cochlear nucleus and the main nuclei in the SOC, known to be involved in sound localization.

Materials and Methods

Animals

Eight fat-tailed dunnarts (Sminthopsis crassicaudata) aged between 12 and 18 months (12–18 g weight) of either sex were used for this study. Precise age was not known but was estimated based on arrival in the animal facilities, weight and time of experimentation. The animals were separately housed in enriched cages containing running discs, rocks and a covered nest. Food (Science Diet Sensitive Stomach Cat Food supplemented with live crickets and mealworms) and water were supplied ad libitum. The vivariums were maintained at 22 °C with a 12-hour Day night cycle. All procedures conformed to NIH guidelines on the use of animals for experimentation (USA) and were approved by the University of Western Australia’s Animal Ethics Committee (RA/3/100/1123).

Auditory brainstem response measurements

The fat-tailed dunnarts were anaesthetised via intraperitoneal injection with ketamine (75 mg/kg) and medetomidine (1 mg/kg). Animals were maintained at near physiological temperature (38 °C) using both a heating pad and an ambient room heater for the entirety of the auditory brainstem response (ABR) recording (60–90 min per animal). ABRs were measured as previously described (Yates et al., 2014). In brief, ABRs were recorded in a sound attenuated room and sound stimuli were generated by custom made Neurosound software (M. Lloyd Cambridge) via a RME DIGI 9636 sound card (96 kHz sampling rate). Average ABRs (n = 400 stimuli) were evoked using pure tone bursts (5 ms duration, 1ms rise-fall-time, rate 10/s), delivered to the animal using a plastic cone attached to a reverse driven 1/4 inch condenser microphone (Brüel and Kjær type 4134). The sound-output was calibrated at the level of the eardrum using a Brüel & Kjær pistonphone (94 dB SPL at 1,000 Hz) (Yates et al., 2014). The acoustic coupler was placed using a surgical microscope to touch the lower edge of the left tragus and was directed towards external auditory meatus. During the course of the experiments, we observed no movement of the animal or auditory coupler.

ABR responses were recorded via an insulated silver-wire electrode inserted subdermally at the vertex. A reference electrode was placed above the left mastoid at the base of the pinna and a ground electrode was inserted into the tail. Differential recordings were made using an AC coupled amplifier (DAM50, World Precision Instruments) with a gain of 1,000x and band pass filtering at (300–3,000 Hz). Average ABR responses were sampled by Powerlab/4ST (AD Instruments) and stored for offline analyses.

ABR thresholds were determined at 4, 8, 16, 24, 32 and 47.5 kHz. In view of the sampling rate of our sound card 47.5 kHz was the maximum frequency tested. Each sound stimulus was presented first at 10 dB attenuation followed by sound intensities decreasing in 10 dB increments until after the disappearance of overt ABR peaks (I and V) in the recording. Upon disappearance of the ABR, the sound intensity was increased in 5 dB steps until the visual reappearance of the peaks in the waveform. Sound stimuli were converted into sound pressure (SPL, re 20 µPa) levels using a Bruel and Kjaer pistonphone (94 dB SPL at 1,000 Hz). ABR traces were analyzed using AxoGraph X V1.5.0 (J. Clements, Australia) and thresholds were determined by visual inspection. ABR threshold was estimated as the lowest intensity where peaks I and V could still be identified. The threshold estimation procedure employed here, was undertaken by three different observers and yielded consistent estimates (less than 5 dB difference).

Histological preparation

Dunnarts were terminally anaesthetised with 0.2 ml euthal (pentobarbitone sodium 170 mg/mL, phenytoin sodium 25 mg/mL). Animals were then perfused with saline (0.9%) followed by paraformaldehyde (4% in 0.1 M phosphate buffered saline, PBS). Regions of brainstem containing auditory nuclei were removed and cryoprotected (30% sucrose in 0.1 M PBS for 24 h) and sectioned at 30 µm using a cryostat (Leica CM1900).

For cresyl violet staining, horizontal sections were washed with PBS for four minutes and then dehydrated in graded ethanol solutions (70%–95%, one minute). Slides were heated in a microwave for 2 min in a 500 mL solution of 95% ethanol and 5% glacial acetic acid (Sigma), followed by rehydration in descending ethanol solutions (95% to 70%, 20 secs each) and washed in PBS for one minute. Sections were then placed in warmed cresyl violet solution (0.5% cresyl violet) for eight minutes. After staining, sections were rapidly exchanged through ascending ethanol solutions (70%–95%, 15 s each) and differentiated at room temperature in 95% ethanol and 5% acetic acid for 5 min. Finally, slides were washed with three 100% ethanol and cleared in xylene. Slides were cover-slipped with DePeX (ProSciTech) mounting media and dried overnight prior to microscopy.

Microscopy and analysis

Images of cresyl violet stained sections were captured using an Olympus DP70 camera and DP Controller (Olympus Corporation, image size 4,080 × 3,072 pixels). High-power micrographs were captured using a Nikon DS-U2/L2 camera with NIS-Elements (Nikon AR 3.0, image size 2,560 × 1,920 pixels). Using standard anatomical markers such as neuronal shape, neuronal density, and somatic alignment, the auditory nuclei (CN and SOC) were identified in the dunnart. Nuclei were observed under low power to determine the area and extent of the nucleus. Images for publication underwent minor adjustments in brightness and contrast.

Results

Auditory brainstem response

A typical ABR was observed in the fat-tailed dunnart (Fig. 1). At moderate to high sound intensities, the ABR showed five distinct peaks within the first 6ms after onset of the tone stimuli. ABRs were evoked at all frequencies tested in this study (between 4 and 47.5 kHz). ABR threshold was estimated as the lowest intensity where peak I and V could still be identified (typical example at 47.5 kHz shown in Fig. 2A). Average thresholds (n = 6 − 8) depicted as audiograms (Fig. 2B) reveal the fat-tailed dunnart ABR is more sensitive (lower thresholds) with increasing frequency. Currently however, it cannot be established whether 47.5 kHz is the most sensitive frequency or if ABR thresholds decline rapidly at higher frequencies.

Figure 1 Characteristic ABR recording from the fat-tailed dunnart (tone burst indicated with black bar below the graph, 47.5 kHz, 5 ms duration, 52 dB SPL).

Grey line represents the background noise from the recording equipment. Main peaks of ABR indicated by roman numerals and accompanied by abbreviated corresponding auditory nuclei. AN, auditory nerve; CV, cochlear nuclei; SOC, superior olivary nuclei; LL, lateral lemniscus; IC, inferior colliculus.

Figure 2 ABR thresholds in fat-tailed dunnart.

(A) ABR recordings at 6 different intensities (42, 32, 22, 12, 7 and 2 dB SPL indicated right of waveforms) in response to a 47.5 kHz tone burst. Waveform corresponding to ABR threshold (disappearance of peaks I and V—shown by asterisks at top) is shown in thick black line. Black bar underneath waveforms indicates duration of tone burst. (B) Audiograms showing ABR thresholds at different frequencies. Individual animal thresholds are shown in grey. (C) Input-output function of the peak I amplitude at 4, 24 and 47.5 kHz. (D) Input-output function of the latency of peak I at 4, 24 and 47.5 kHz. Each data point shows mean ± SEM. N.B. in (C) and (D) some of the points at very low sound intensity are the values derived from one or two animals.

In agreement with the known characteristics of ABR responses, peak I amplitudes increased with increasing sound intensity (Fig. 2C). Similarly, increasing sound intensities resulted in a shortening of ABR latencies (data for 4, 24 and 47.5 kHz shown in Fig. 2D).

Histological analysis

The cochlear nerve root and cochlear nuclei

Similar to other known marsupial species such as the brush-tailed possum and quoll, the cochlear nuclei (CN) reside medial to the restiform body (rb in Figs. 3A and 3C). The ventral cochlear nucleus (VCN) as a whole is clearly identifiable in the dunnart (Figs. 3C and 3H) with round small closely packed cells of the anteroventral cochlear nucleus (AVCN) in rostral levels to the dorsal cochlear nucleus (DCN). A sparsely populated posteroventral cochlear nucleus (PVCN) containing large nuclei was observed in more caudal sections containing the other cochlear nuclei (Figs. 3C–3F).

Figure 3 Overview of the fat-tailed dunnart auditory brainstem.

Nissl staining of transverse sections reveals prominent auditory nerve root nucleus and cochlear nuclei. Images are organised caudal to rostral. The dorsal cochlear nucleus resides medio-dorsal to the restiform body in the caudal regions (shown in A, with high power image in B). (C and D): More rostrally the ventral cochlear nucleus shows prominently as well. (D), (E), and (F): further rostral the trilaminar arrangement of the dorsal cochlear nucleus is clearly visible (F) as well as the cochlear nerve root nucleus (G). At more rostral level (H) the ventral cochlear nucleus shows a separation between posteroventral and anteroventral cochlear nucleus. Scale bars are 500 µm in A, C, E, H and 200 µm in B, D and F, G. Distance between A and C: 240 µm, between C and E 90 µm, and between E and H 210 µm. Section coordinates in (H) are for all sections. Abbreviations: cnr, cochlear nerve root; cb, cerebellum; dcn, dorsal cochlear nucleus; fn, facial nucleus; rb, restiform body, avcn, anteroventral cochlear nucleus; pvcn, posteroventral cochlear nucleus; in F: I–molecular layer II–fusiform layer III–polymorphic layer.

On gross appearance, the dunnart DCN is a trigonal shaped nucleus. In more caudal sections, a prominent tri-laminar DCN could clearly be subdivided into a molecular (I in Fig. 3F), fusiform layer (II in Fig. 3F) and polymorphic layers (III in Fig. 3F). The DCN was bounded laterally by the small cell cap layer (scc, Figs. 3D and 3F).

Briefly. the dunnart also shows a clearly defined cochlear nerve root nucleus (CNR) (Figs. 3E and 3G), consisting of large neurons clustered within the passing nerve fascicles.

The superior olivary complex nuclei

The nuclei of the superior olivary complex (SOC) in the dunnart closely resembled their anatomical correlates found in eutherian mammals. Of the three principal SOC nuclei lateral superior olive (LSO), medial superior olive (MSO), and the medial nucleus of the trapezoid body (MNTB), the most prominent and distinguishable nucleus in the dunnart was the MNTB (Figs. 4A and 4B). The MNTB occupied a familiar position within the brainstem and the cells of the MNTB were not densely packed presumably due to their location within the passing trapezoid body projection (see Fig. 4B). A small MSO (typically observed within one to two histological sections) was observed as a linear cluster of pleiomorphic cells aligned along a dorsal-ventral axis (Figs. 4A–4C).

Figure 4 The superior olivary complex (SOC) nuclei in the fat-tailed dunnart.

The four main nuclei evident include the medial nucleus of the trapezoid body (MNTB) (A and B) residing within the fibres of the trapezoid body. The superior paraolivary nucleus (SPN) is located dorsomedial to the MNTB. Located laterally to the MNTB and SPN is the linear medial superior olive (MSO) (A, B with outline in C). The lateral superior olive (LSO) (outline in C) can be seen lateral to the MSO and contains a marginal (lso-m) and core (lso-c) regions (outlines in A, C). The boundary of the LSO shown in C is tentative and derive from alignment of neuronal somata. Micrographs are taken at 2× (A) and 10× (B). Section coordinates in (C) are for all sections. Scale bars denote 1 mm in A and 200 µm in B, C. Abbreviations: lso, lateral superior olive; mntb, medial nucleus of the trapezoid body; mso, medial superior olive; spn, superior paraolivary nucleus; tb, trapezoid body.

The lateral superior olive (LSO) of the dunnart was not as well defined as found in similarly sized eutherian species (Fig. 4C). Despite this, the LSO was observed as a round nucleus located near the latero-ventral surface of the brainstem in transverse sections often containing the MNTB. The LSO could be subdivided into densely stained elongated cells in more marginal areas (Fig. 4C, lso(m)), whereas lightly stained bipolar nuclei were found to occupy more core or central locations (Fig. 4C, lso(c)).

A nucleus corresponding to superior paraolivary nucleus (SPN, Figs. 4A–4C), was observed residing dorsomedially between the MNTB and the MSO. Somata in this region contained large densely stained multipolar cells with no clear systematic orientation.

Discussion

Here we characterise some of the anatomical and electrophysiological features of the ascending auditory pathway in the fat-tailed dunnart. With the exception of Aitkin (1998), there has been very little characterisation of the dunnart auditory system, therefore we sought to establish normative values of the fat-tailed dunnart auditory system. In addition to identifying common auditory nuclei, we found that the anesthetised fat-tailed dunnart auditory system is remarkably sensitive to high frequency stimuli.

The ABR represents the average response to repetitive sound stimuli of neuronal populations in the auditory pathway. Waveform analysis of the ABR revealed 5 definite peaks (Reimer, 1996) with short latency, corresponding to the action-potential volleys from the auditory nerve through to inferior colliculus (Liu, Hill & Mark, 2001). In the current study, not only were we still able to evoke ABR responses to high frequency stimuli (47.5 kHz), but ABR thresholds improved at high frequencies. These ABR findings are puzzling and present a contrast to the only previously published data from a dunnart species (Sminthopsis macroaura), which displayed a frequency range of 1–40 kHz and a minimum, or best threshold at 10 kHz (Aitkin, 1998). However, this study was limited by low animal numbers (n = 2) and lack of detail in the methodology, making it unclear whether 40 kHz was the highest frequency attempted.

Nonetheless, high frequency sensitivity is quite common in small non-echolocating mammals such as the leaf-eared mouse and spiny mouse (Heffner, Koay & Heffner, 2001). In fact, upon closer inspection of cochlear and ABR audiograms taken from several rodent species including the mouse (Mus musculus), a second local minimum is present (20–30 dB SPL) at around 50 kHz (Ehret, 1976; Heffner, Koay & Heffner, 2001), and similarly, secondary local minima are also found in echo-locating mammals (∼15 dB SPL at >45 kHz) (Koay, Heffner & Heffner, 1998).

With the exception of the cat (Felis catus), animals with smaller head sizes have small functional interaural distances and tend to have higher audible frequencies (Heffner, Koay & Heffner, 2001; Koay, Heffner & Heffner, 1998). In agreement with this, another marsupial, the northern quoll (Dasyurus hallucatus) which is larger than the dunnart (adults 400 g, 5 cm snout-ear), is most sensitive at 10 kHz (10 dB SPL) with rapid loss of sensitivities at 40 kHz (50–80 dB SPL) (Aitkin, Nelson & Shepherd, 1994; Oakwood, 2002). Similarly, the Brazilian short-tailed opossum (Monodelphis domestica) also a marsupial larger than the fat-tailed dunnart (rat-size) shows best thresholds between 8 and 12 kHz (20 dB SPL) and an upper audible frequency limit of 60 kHz (Reimer, 1995). Therefore, given its small size (12–18 g), the high frequency sensitivity observed in the fat-tailed dunnart may be in line with its size, but conflicts with the limited data from the stripe-faced dunnart (Aitkin, 1998), which is of similar size. Therefore, we cannot exclude the possibility that this audiogram of the fat-tailed dunnart represents a specific adaptation to its auditory environment, in line with the specific adaptation found in its visual system (Cowing et al., 2008; Ebeling, Natoli & Hemmi, 2010). The reasons for such specialised adaptations within its sensory system remain unclear. As discussed in Ebeling et al. it may represent specific adaptations to the visual and auditory ecology or, alternatively, adaptations in early ancestors (Ebeling, Natoli & Hemmi, 2010).

The anatomy of the auditory brainstem in the fat-tailed dunnart reveals a similar pattern of auditory nuclei as reported previously across a range of marsupials (Aitkin, 1998). The CNR is present in many small marsupials including the yellow-bellied glider (Petaurus australis), Northern quoll (Aitkin, Byers & Nelson, 1986) but also in muridae (López et al., 1993; Merchan et al., 1988). While neurons in the CNR nucleus are considered as an extension of the ventral cochlear nucleus (Osen et al., 1991), it projects to motor components of the pontine reticular and facial nuclei (Lopez et al., 1999). Although few in number, neurons in the CNR nucleus in the rat respond to sound and thus likely represent an initial auditory nucleus (Sinex, Lopez & Warr, 2001). Given its early position within the auditory pathway, sensitivity to sound, and efferent projections to the pontine motor nuclei, the CNR nucleus is thought to play a role in sensorimotor control of acoustic startle responses (Lee et al., 1996).

The auditory cochlear nuclei in the dunnart were similar in location to other marsupial species studied such as the brush-tailed possum (Trichosurus vulpecula) (Aitkin & Kenyon, 1981), multiple glider species (Aitkin, 1996), and northern quoll (Aitkin, Byers & Nelson, 1986). Also in agreement with other marsupials, the fat-tailed dunnart’s trilaminar DCN was larger than the VCN (Aitkin, 1996; Aitkin, 1998). Despite widespread variation across mammalian species (Glendenning & Masterton, 1998; Illing, Kraus & Michler, 2000), the organisation of the SOC was again largely consistent with previous reports. In common laboratory rodents, the three main SOC (LSO, MSO and MNTB) as well as the SPN, are known targets of the cochlear nuclei and it is likely that a similar connectivity exists in marsupials (Aitkin, Byers & Nelson, 1986; Bazwinsky-Wutschke et al., 2016; Schofield, 1995). The presence of the MSO is not surprising as it is known to persist in almost all mammalian species including the mouse (Fischl et al., 2016; Ollo & Schwartz, 1979). Furthermore, the location and appearance of the MSO as a linear nucleus in the dunnart is consistent with previous reports in arboreal marsupials (Aitkin, 1996). The MSO is involved in detecting interaural timing differences related to sound localization of lower frequencies (Grothe & Sanes, 1994). Therefore, it is likely that the functional role of the MSO in these small animals with high frequency sensitivity is relatively limited (Grothe & Pecka, 2014) and hence its small size in the fat-tailed dunnart is as expected. The LSO and MNTB, involved in detection of higher frequencies based on interaural level differences (Caird & Klinke, 1983; Grothe & Koch, 2011) were both present in the fat-tailed dunnart in line with its high frequency sensitivity.

The relative size of the MNTB is known to vary between species, its relative size being about 5% of the subcortical auditory system in kangaroo rat and less than 1% in humans (Glendenning & Masterton, 1998). In addition, a study by Hilbig et al. comparing different primates, showed a marked reduction in MNTB size from macaque to human (Hilbig et al., 2009). The MNTB in the fat-tailed dunnart was clearly distinguishable with large neurons comparable to the anatomy in rat (Reuss et al., 1999). Taken together, the presence of the large well defined MNTB, which is known to provide powerful and precisely-timed glycinergic inhibition to the ipsilateral LSO, MSO and SPN (Adams & Mugnaini, 1990; Sanes & Friauf, 2000), completes known anatomical circuits involved with sound-localisation in many eutherian species (Kapfer et al., 2002; Sanes & Friauf, 2000; Tollin, 2003).

The LSO is often described as an S-shaped or horseshoe shaped nucleus in many species such as guinea pig, cat and gerbil (Grothe & Park, 2000). In contrast to the mouse (Ollo & Schwartz, 1979), a distinct shape could not be observed in our histological material. Rather the LSO boundary in the dunnart remained diffuse, in line with the description of Aitkin (1996) in some arboreal marsupials (Aitkin, 1996). Despite the lack of a clear boundary, the LSO did contain subdivisions (marginal and core) which has been reported previously by Willard & Martin (1983) in the opossum (Willard & Martin, 1983).

While the presence of CN and SOC in the dunnart suggests an ability to process incoming auditory information particularly in terms of sound localisation, further investigations into the synaptic morphology, neurochemistry and electrophysiology would further help to refine our understanding of the roles these nuclei play within the dunnart and their environment.

Conclusions

Here we show that the fat-tailed dunnart is an animal species that displays a remarkable high frequency sensitivity. In addition, the auditory brainstem nuclei reveal a large and well developed CN as well as a MNTB. These nuclei are important in early binaural auditory processing and sound localisation, and their presence in the dunnart suggests similar processing capabilities. In addition to extending the ABR audiograms to higher frequencies, it would be of immediate interest to determine how the hearing sensitivities correspond to species specific communication as well as predator/prey detection and avoidance (Aitkin, Nelson & Shepherd, 1994). In light of recent reports on the role of the DCN in the analysis of vocalisations (Roberts & Portfors, 2015), it would be of interest to determine if the DCN performs a similar role in the marsupial.

Supplemental Information

Data S1 Raw ABR data files used to generate Figs. 1 and 2

The data contained in this file represent the individual ABR responses for each fat-tailed dunnart. Each dunnart was tested at multiple frequencies and sound-intensities which are also shown at the top of each column.

Click here for additional data file.

Additional Information and Declarations

Competing Interests

Author Contributions

Animal Ethics

Data Availability

Jennifer Rodger is an Academic Editor for PeerJ.

Andrew Garrett and Nathanael J. Yates conceived and designed the experiments, performed the experiments, analyzed the data, prepared figures and/or tables, authored or reviewed drafts of the paper, approved the final draft.

Virginia Lannigan performed the experiments, analyzed the data, prepared figures and/or tables, approved the final draft.

Jennifer Rodger and Wilhelmina Mulders conceived and designed the experiments, performed the experiments, analyzed the data, contributed reagents/materials/analysis tools, prepared figures and/or tables, authored or reviewed drafts of the paper, approved the final draft.

The following information was supplied relating to ethical approvals (i.e., approving body and any reference numbers):

The University of Western Australia’s Animal Ethics Committee approved this research (RA/3/100/1123).

The following information was supplied regarding data availability:

The raw measurements are available in the Supplemental File. The raw data shows the ABR responses from each Dunnart recorded at the corresponding frequency and sound-pressure level.

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
