# Peer review of "Physiological and anatomical investigation of the auditory brainstem in the Fat-tailed dunnart (Sminthopsis crassicaudata)"

_PeerJ, doi:10.7717/peerj.7773_

## Round 0.1 · original submission · Minor Revisions

As you will see, the reviewers provided very clear and helpful suggestions for improvements to the manuscript. I believe that the majority of the concerns raised can be addressed with a combination of text revisions and minor edits to the figures. However, Reviewer 3's comments regarding the ABR, and in particular, the estimation of thresholds by visual inspection, will require careful attention. Please indicate by formatting or highlighting the revised text in the re-submitted manuscript, and thank you again for sending your work to PeerJ.

·

Basic reporting

no comment

Experimental design

no comment

Validity of the findings

no comment

Additional comments

The article by Garrett et al. describes auditory brainstem responses (ABR) in the anaesthetized fat-tailed dunnart, with a surprising finding of lower thresholds for higher frequencies towards 50kHz. Unfortunately, higher frequencies are not examined due to technical limitations, and therefore peak sensitivities could not be tested. Although a more complete characterization would be needed to answer this, the present article discusses these limitations appropriately, and within a well-framed comparative context. Next, the authors present a detailed cytoarchitectural description of brainstem auditory nuclei in this species using Nissl staining. In my opinion, this is a timely and well-written article that describes some very basic features of dunnart auditory neurobiology, particularly relevant due to the scarcity of data in dasyurid marsupials.

I only have minor comments for the authors to address before proceeding to publication:

- Use italics or underline for all species’ scientific names (lines 69-72).
- There’s a space missing in lines 81 and 104.
- Do not use capital letters for common names or reagents (e.g. fat-tailed dunnart in title; ketamine, cresyl violet, etc)
- Fig. 1 title says ‘d’at-tailed
- Figs 3 and 4 should clearly indicate the plane of section in the result section and figure legends (at least coordinates such as 'lateral', 'anterior' to ease spatial comparisons). Including a schematics of the approximate region where sections were taken would be great in both Fig. 3 and 4.
- Improve the labelling of the nuclei of interest in Figs. 3 and 4, for example by outlining all regions and/or placing the lettering at the centre of each structure, or outside with a line pointing to them, to improve clarity and flow with the text.

Reviewer 2 ·

Basic reporting

The present study investigated the auditory system of the marsupial fat-tailed Dunnart (Sminthopsis crassicaudata), because auditory brainstem of this species is not investigated in detail so far.
Auditory brainstem responses were recorded showing similar response to other mammalian species, however, the dunnart is most sensitive to higher frequencies increasing up to 47.5 kHz. Morphological investigation of auditory brainstem by use of Nissl-staining revealed auditory nuclei (cochlear nerve root nucleus, cochlear nucleus subdivisions and nuclei of the superior olivary complex) comparable to that in other mammalian/marsupial species.
The manuscript in general is well written, however a more detailed anatomical/histochemical analysis of auditory brainstem nuclei would be desirable. The manuscript should be improved as suggested in general comments for the author.

Experimental design

The research question is clearly defined and would be of interest to know whether auditory brainstem organization in fat-tailed Dunnart is similar or different to that in eutherian or other metatherian species with respect to range of frequency hearing.
Methods are described with sufficient Information to be reproducible. Still, histological analysis of auditory brainstem nuclei can be improved by a more detailed Investigation of auditory nuclei (subdivisions, size, extent, neuron types, myeloarchitecture).

Validity of the findings

Validity of findings is given, but can be improved by a more detailed morphological characterization of auditory nuclei (cochlear nucleus with each subregion, single nuclei of the superior olilvary complex) supporting physiological data.

Additional comments

Introduction
Page 3-4: The auditory brainstem nuclei of marsupials display distinct features compared to eutherian mammals, this should be more clarified in the introduction.
Results
Page 8, line 199: A more sparsely populated PVCN containing larger nuclei was observed in more caudally located sections. Please clarify this statement, what is think about “containing larger nuclei” in more caudally located sections (Figure 3c,d,e,f)?
Page 8, line 195, line 202, line 209: Literature should not to be included into the results, but are a part of discussion.
Page 8: line 194-200: In marsupials the partition of the CN is somewhat different from that of the eutherian mammals: anterolateral and posteromedial AVCN as well as lateral PVCN and octopus cell area are named and should be also identified in the here investigated marsupial.
Page 8, line 201: At the beginning of the DCN, the DCN appears not to be tri-laminar (Figure 3b, d)?! Clearly lamination was only seen in Fig. 3f. The correct designations of the DCN layers are as follows beginning with the outermost layer: molecular layer, fusiform layer and polymorphic layer (see also Bazwinsky-Wutschke et al. 2008).
Page 9, line 221, line 223: Avoid the involvement of citations in the result section.
The description of the LSO is very sparsely. According to other marsupials, the drop-shaped LSO can be subdivided into a central and marginal region (see also Willard and Martin 1983). Do you have seen the SPN (superior paraolivary nucleus) in the SOC?
To improve and to deepen the histological analysis, in general the rostro-caudal extent of each auditory brainstem nucleus can be analyzed and features described.
Discussion
Page 11: The MNTB-LSO system should be more discussed with the background that the MNTB is the major source of acoustically evoked glycinergic inhibition within SOC and a key nucleus in the sound localization (transforming the bushy cell excitatory input into inhibitory output acting on the ipsilateral LSO, MSO and SPN).

Reviewer 3 ·

Basic reporting

The article was well written, clear and concise.
Literature citations were generally appropriate.
Article was structured well.

Experimental design

Experimental design followed prior methods, so I have no issues.
Research question was well defined.

Validity of the findings

The methods used were tried-and-true. Thus, I have confidence that the findings and conclusions are valid.

Additional comments

What follows are just some general comments and questions that the authors might consider.
Ln 105. Was sex not known?
Ln 114. Some additional details regarding the ABR measurements would be helpful. How was the system calibrated? How was SPL of the stimulus measured? In the ear canal? Or simply the output of the earphone?
Ln 143. Visual estimation of ABR threshold can be quite variable. To give the reader some sense of what was done, could you provide the estimate of the threshold for the example ABRs shown in Figure 2a?

---

## Round 0.2 · accepted · Accept

Thank you for your attention to the comments from the reviewers. While it is certainly not worth holding up progress of the manuscript, I believe that there is a minor editing error in line 72, p.3 of the revised manuscript- "...largely consistent within eutherian mammals..." should be changed to "...largely consistent with eutherian mammals...".
Thank you again for submitting your work to PEERJ, and I look forward to seeing the final published version of this manuscript.